# Comparison of the Effectiveness and Safety of Heterologous Booster Doses with Homologous Booster Doses for SARS-CoV-2 Vaccines: A Systematic Review and Meta-Analysis

**DOI:** 10.3390/ijerph191710752

**Published:** 2022-08-29

**Authors:** Jie Deng, Yirui Ma, Qiao Liu, Min Du, Min Liu, Jue Liu

**Affiliations:** 1Department of Epidemiology and Biostatistics, School of Public Health, Peking University, Beijing 100191, China; 2Institute for Global Health and Development, Peking University, Beijing 100191, China

**Keywords:** SARS-CoV-2, COVID-19 vaccine, heterologous booster dose, homologous booster dose, effectiveness, safety, immunogenicity

## Abstract

As vaccine resources were distributed unevenly worldwide, sometimes there might have been shortages or delays in vaccine supply; therefore, considering the use of heterogeneous booster doses for Coronavirus disease 2019 (COVID-19) might be an alternative strategy. Therefore, we aimed to review the data available to evaluate and compare the effectiveness and safety of heterologous booster doses with homologous booster doses for severe acute respiratory syndrome coronavirus 2 (SARS-CoV-2) vaccines. We searched relevant studies up to 27 April 2022. Random-effects inverse variance models were used to evaluate the vaccine effectiveness (VE) and its 95% confidence interval (CI) of COVID-19 outcomes and odds ratio (OR) and its CI of safety events. The Newcastle–Ottawa quality assessment scale and Cochrane Collaboration’s tool were used to assess the quality of the included cohort studies. A total of 23 studies involving 1,726,506 inoculation cases of homologous booster dose and 5,343,580 inoculation cases of heterologous booster dose was included. The VE of heterologous booster for the prevention of SARS-CoV-2 infection (VE_heterologous_ = 96.10%, VE_homologous_ = 84.00%), symptomatic COVID-19 (VE_heterologous_ = 56.80%, VE_homologous_ = 17.30%), and COVID-19-related hospital admissions (VE_heterologous_ = 97.40%, VE_homologous_ = 93.20%) was higher than homologous booster. Compared with homologous booster group, there was a higher risk of fever (OR = 1.930, 95% CI, 1.199–3.107), myalgia (OR = 1.825, 95% CI, 1.079–3.089), and malaise or fatigue (OR = 1.745, 95% CI, 1.047–2.906) within 7 days after boosting, and a higher risk of malaise or fatigue (OR = 4.140, 95% CI, 1.729–9.916) within 28 days after boosting in heterologous booster group. Compared with homologous booster group, geometric mean neutralizing titers (GMTs) of neutralizing antibody for different SARS-CoV-2 variants and response rate of antibody and gama interferon were higher in heterologous booster group. Our findings suggested that both homologous and heterologous COVID-19 booster doses had great effectiveness, immunogenicity, and acceptable safety, and a heterologous booster dose was more effective, which would help make appropriate public health decisions and reduce public hesitancy in vaccination.

## 1. Introduction

Caused by severe acute respiratory syndrome coronavirus 2 (SARS-CoV-2), the pandemic of coronavirus disease 2019 (COVID-19) has posed an extraordinary threat and burden of disease to global public health [1]. According to the World Health Organization (WHO) COVID-19 dashboard, as of 8 July 2022, there have been more than 551.2 million confirmed cases of COVID-19, including more than 6.34 million deaths, globally [2]. Vaccination against SARS-CoV-2 is still considered a key measure to prevent infection, serious illness, hospitalization, and death, as well as control the COVID-19 pandemic [3]. As of 10 July 2022, a total of more than 12.14 billion doses of COVID-19 vaccine have been administered globally, and 5.54 million are now administered each day [2,4]. A total of 66.7% of the world population has received at least one dose of COVID-19 vaccine; however, only 20.2% of people in low-income countries have received at least one dose [4]. In 2021, WHO set the target for 70% global vaccination coverage by mid-2022 [5,6]. As of June 2022, only 58 of WHO’s 194 member states had reached the 70% target, and in low-income countries, just 37% of healthcare workers had received a complete course of primary vaccination [2,5]. Equity and availability of vaccine resources, therefore, still remain a key issue in achieving the global vaccine coverage rate target.

Emerging studies consistently showed that the initial vaccine effectiveness (VE) for the prevention of SARS-CoV-2 infection and COVID-19 disease decreased over time since vaccination [7]. Therefore, WHO proposed to give booster doses to vaccinated people who have completed the primary vaccine series to restore and prolong the VE when immunity and clinical protection rates fell below levels considered adequate in that population [8,9]. According to the latest WHO Strategic Advisory Group of Experts on Immunization (SAGE) roadmap for prioritizing use of COVID-19 vaccine, WHO recommended that countries with moderate-to-high rates of primary series coverage in higher priority-use groups should usually prioritize available resources to first achieve high booster dose coverage rates in higher priority-use groups before offering vaccine doses to lower priority-use groups [10]. However, vaccine resources were unevenly distributed worldwide, with extremely low vaccination rates in some low-income countries [5]. Additionally, sometimes there might be shortages or delays in vaccine supply [11]. Therefore, considering the use of heterogeneous booster doses for SARS-CoV-2 might be an alternative strategy to the homologous to alleviate these problems to some extent and promote the equity and rationalization of vaccine allocation. It was important to evaluate different COVID-19 booster vaccination strategies, which would help public policy decisions and reduce vaccination hesitancy.

WHO issued interim guidance summarizing existing evidence of heterologous primary and boosting SARS-CoV-2 vaccine and setting vaccination schedules on 16 December 2021 [12] and also stated that it was safe and effective to receive a different third dose of COVID-19 vaccine [3]. Au, WY. et al.’s systematic review has showed that heterologous and homologous three dose regimens work comparably well in preventing COVID-19 infections, even against different variants but remained uncertain in COVID-19-related death. The VE of two dose adenovirus vector vaccines with one mRNA vaccine in the prevention of COVID-19 infection was 88%, while the most effective regimen with three dose mRNA vaccine was 96% [13]. Li JX et al. found that a heterologous booster vaccine with an orally administered aerosolized Ad5-nCoV was safe and highly immunogenic in Chinese adults who have previously received two doses of CoronaVac as the primary series vaccination [14]; furthermore, heterologous boosting with Convidecia following initial vaccination with CoronaVac was safe and more immunogenic than homologous boosting [15], which was similar to previous studies that suggested the safety and highly immunogenicity of heterologous booster strategy [16,17,18]. Atmar, R. L. et al.’s study showed that more than half the recipients reported having injection-site pain, malaise, headache, or myalgia but were acceptable [19].

Since the advent of COVID-19 vaccine, as of 19 May 2022, WHO has validated 11 vaccines for COVID-19 emergency use listing (EUL) [20,21,22]. Inconsistent procurement of COVID-19 vaccines and limited vaccine supplies prevented some types of vaccines from being used clinically. Even though there were more and more clinical trial studies of COVID-19 boosting vaccination strategy, evidence for the effectiveness, immunogenicity, and safety of heterologous booster doses of COVID-19 vaccine remained partial and incomplete [12]. Timely evaluation of different booster vaccination strategies would help in making appropriate public health decisions, reducing public hesitancy in vaccination, and promoting rational allocation of COVID-19 vaccine resources. We aimed to compare the effectiveness, immunogenicity, and safety of heterologous booster doses with homologous booster doses for COVID-19 vaccines based on the real world and conducted a systematic review and meta-analysis to provide an evidence-based basis for the COVID-19 boosting vaccination strategy.

## 2. Methods

### 2.1. Search Strategy and Selection Criteria

We searched studies published up to 27 April 2022 without language restrictions in PubMed, EMBASE, Web of Science, Science Direct, and Cochrane Library databases using the following search terms: (SARS-CoV-2 vaccine) OR (COVID-19 vaccine)) AND (booster dose)) AND (heterologous OR homologous).

We used EndNoteX8.2 (Thomson Research Soft, Stanford, CA, USA) to manage records. This study was strictly performed according to the preferred reporting items for systematic reviews and meta-analyses (PRISMA in the Appendix A) [23]. This study was registered on PROSPERO (CRD42022328792).

We included studies that examined the effectiveness and safety of heterologous booster doses with homologous booster doses for SARS-CoV-2 vaccines. The following studies will be excluded: (1) irrelevant to the subject of the meta-analysis, such as studies that did not use SARS-CoV-2 vaccination as the exposure; (2) insufficient data to calculate the rate for the effectiveness and safety of SARS-CoV-2 vaccines; (3) duplicate studies or overlapping participants; (4) qualitative researches, reviews, editorials, conference papers, case reports, or animal experiments; and (5) studies that did not clarify the identification of COVID-19. For example, the confirmed diagnosis of COVID-19 via reverse-transcription polymerase chain reaction (rt-PCR) test, serologic test, or other means was not mentioned in the text.

Studies were identified by two investigators (DJ and MYR) independently following the criteria above, while discrepancies were solved by consensus or with a third investigator (LQ).

### 2.2. Data Extraction

The following data will be extracted: (1) basic information of the studies, including first author, published time, and article type (preprint article or published article), study design (e.g., cohort study, case-control study, or randomized controlled trial (RCTs)), and location where the study was conducted; (2) characteristics of the study population, including population sizes, age, sex ratio (female/male), vaccination status and vaccine type for prime regime and booster, and underlying disease; (3) effectiveness of heterologous booster and homologous booster for SARS-Cov-2 vaccines: vaccine effectiveness (VE) for prevention of adverse outcomes, including SARS-CoV-2 infection, symptomatic COVID-19, emergency department and urgent care admissions, COVID-19 hospital admissions, COVID-19 ICU admissions, and COVID-19-related death; (4) safety of heterologous booster and homologous booster for SARS-CoV-2 vaccines: number or proportion of cases that had systematic and injection site adverse reaction after vaccination of booster and follow-up period; and (5) immunogenicity of heterologous booster and homologous booster for SARS-Cov-2 vaccines: geometric mean neutralizing titers (GMTs) against different SARS-Cov-2 variants and response rate of antibody and interferon.

Data extraction was conducted by two investigators (DJ and MYR) independently following the criteria above, while discrepancies were solved by consensus or with a third investigator (LQ).

### 2.3. Quality Assessment

We used the Newcastle-Ottawa quality assessment scale to evaluate the risk of bias of cohort studies and case-control studies and Cochrane Collaboration’s tool for RCTs. Cohort studies and case-control studies were classified as having low (≥7 stars), moderate (5–6 stars), and high risk of bias (≤4 stars) with an overall quality score of 9 stars. RCTs were classified as low, unclear, or high risk of bias from comprehensive evaluation from 7 dimensions, including random sequence generation, allocation concealment, blinding of participants and personnel, incomplete outcome data, selective reporting, and other bias.

Quality assessment was conducted by two investigators (DJ and MYR) independently, while discrepancies were solved by consensus or with a third investigator (LQ).

### 2.4. Data Synthesis and Statistical Analysis

We performed a meta-analysis to estimate the VE and its 95% confidence interval (CI) against adverse outcomes and odds ratio (OR) and its 95% CI of the safety events after vaccination of heterologous booster and homologous booster for SARS-CoV-2 vaccines. We also estimated the standard mean difference (SMD) and its 95% CI of GMTs against different SARS-CoV-2 variants and OR and its 95% CI of response rate for antibody and interferon. We performed subgroup analyses by article type (preprint article or published article), study design (cohort study, case-control study, or RCTs), and immune condition. For example, inactivated vaccine ×2 + mRNA vaccine ×1 meant 2 primary doses of inactivated vaccines and a booster dose of mRNA vaccine.

Random-effects or fixed-effects models were used to pool the rates and adjusted estimates across studies separately, based on the heterogeneity between estimates (I^2^). Fixed-effects models would be used if I^2^ ≤ 50%, which represents low to moderate heterogeneity, and random-effects models would be used if I^2^ ≥ 50%, representing substantial heterogeneity. The D-L method was used to estimate the tau square in cases of random-effects models. Publication bias was assessed by Harbord’s modified test. All analyses used Stata version 16.0 (Stata Corp, College Station, TX, USA).

## 3. Results

### 3.1. Basic Characteristics

In the initial literature research, 779 potential articles were identified up to 27 April 2022, including 424 published articles (118 in PubMed, 153 in Embase, 153 in Web of Science) and 355 preprint articles (140 in Biorxiv, 215 in Medrxiv). A total of 256 duplicates was excluded. After reading the titles and abstracts, 402 articles were excluded based on the inclusion and exclusion criteria. Among the 121 studies under full-text review, 98 studies were excluded. Eventually, 23 studies (including 13 published articles and 10 preprints) were included in this meta-analysis based on the inclusion criteria [14,24,25,26,27,28,29,30,31,32,33,34,35,36,37,38,39,40,41,42,43,44,45]. The literature retrieval flow chart is shown in Figure 1.

Of the 23 studies included, 9 were cohort studies, 11 were clinical trials, 2 were case-control studies, and 1 could not be determined. The included studies described or compared the effectiveness and safety of heterologous booster doses with homologous booster doses for SARS-CoV-2 vaccines and involved 1,726,506 inoculation cases of homologous booster dose and 5,343,580 inoculation cases of heterologous booster dose. The characteristics of the included studies are shown in Appendix A.

### 3.2. Comparison of Safety between Heterologous Booster and Homologous Booster

We evaluated the systematic adverse reactions with 7 and 28 days after boosting, as well as the injection site adverse reactions with 7 days after boosting. Compared with homologous booster group, there was a higher risk of fever (OR = 1.930, 95% CI, 1.199–3.107), myalgia (OR = 1.825, 95% CI, 1.079–3.089), and malaise or fatigue (OR = 1.745, 95% CI, 1.047–2.906) within 7 days after boosting, and a higher risk of malaise or fatigue (OR = 4.140, 95% CI, 1.729–9.916) within 28 days after boosting in the heterologous booster group (*p*-value < 0.05). No differences in other systematic or injection site adverse reactions were observed between the homologous and heterologous booster groups. The analysis results are shown in Table 1.

### 3.3. Comparison of Immunogenicity between Heterologous Booster and Homologous Booster

We evaluated the GMTs of neutralizing antibody for different SARS-CoV-2 variants and response rate of antibody and gama interferon. Compared with homologous booster group, GMTs of anti-RBD IgG (SMD = 1.244, 95% CI, 0.900–1.588) at 14 days were higher after boosting in the heterologous booster group. Compared with the homologous booster group, GMTs of anti-wild-type neutralizing antibody at 14 and 28 days after boosting (SMD = 1.028, 95% CI, 0.654–1.402; SMD = 0.967, 95% CI, 0.571–1.363), and GMTs of anti-Delta neutralizing antibody at 14 days after boosting (SMD = 0.833, 95% CI, 0.509–1.157) were higher in the heterologous booster group (*p*-value < 0.05).

Compared with the homologous booster group, there was higher response rate of anti-spike IgG at 28 days after boosting (OR = 5.536, 95% CI, 2.738–11.195) and gama interferon at 14 days after boosting in patients who were negative in baseline (OR = 16.3, 95% CI, 3.439–77.272) in the heterologous booster group (*p*-value < 0.05). No differences for response rate of neutralizing antibody nearly one month after boosting were observed between the homologous and the heterologous booster groups. The analysis results are shown in Table 2.

### 3.4. Comparison of Vaccine Effectiveness between Heterologous Booster and Homologous Booster

In the homologous booster group, the pooled VE was 84.00% (95% CI, 82.8–85.1%) for the prevention of SARS-CoV-2 infection, 17.30% (95% CI, 14.4–20.2%) for the prevention of symptomatic COVID-19, 82.70% (95% CI, 81.7–83.7%) for the prevention of emergency department and urgent care admissions, 93.20% (95% CI, 92.4–94.0%) for the prevention of COVID-19 hospital admissions.

In the heterologous booster group, the pooled VE was 96.10% (95% CI, 95.8–96.3%) for the prevention of SARS-CoV-2 infection, 56.80% (95% CI, 56.3–57.4%) for the prevention of symptomatic COVID-19, 97.40% (95% CI, 97.1–97.7%) for the prevention of COVID-19 hospital admissions, 98.70% (95% CI, 98.4–99.1%) for the prevention of COVID-19 ICU admissions and 98.00% (95% CI, 97.4–98.6%) for the prevention of COVID-19-related death.

The analysis results are shown in Table 3, which suggests that the vaccine effectiveness of the heterologous booster for the prevention of SARS-CoV-2 infection, symptomatic COVID-19, and COVID-19-related hospital admissions was higher than the homologous booster.

### 3.5. Quality Evaluation and Publication Bias

We evaluated the quality of the included cohort studies and case-control studies according to the Newcastle-Ottawa quality assessment scale, all of them were of good quality and had a low risk of bias (≥7 stars), as shown in Table 4 and Table 5. The Cochrane Collaboration’s tool was used to assess the risk of bias of the included RCTs, and the results suggested that most of the studies were at low risk of bias, followed by unclear risk of bias, as shown in Figure 2.

**Table 4 ijerph-19-10752-t004:** Risk of bias and quality assessment by Newcastle-Ottawa quality assessment scale (NOS) of the included cohort studies.

First Author (Published Time)	Selection				Comparability	Outcome			Number of Stars	Risk of Bias
	Representativeness of the Exposed Cohort	Selection of the Non-Exposed Cohort	Ascertainment of Exposure	Demonstration that Outcome of Interest Was Not Present at Start of Study	Comparability of Cohorts on the Basis of the Design or Analysis	Assessment of Outcome	Was Follow-Up Long Enough for Outcomes to Occur	Adequacy of Follow-Up of Cohorts		
Ai, J. et al., 2022 [25]	0	0	1	1	2	1	1	1	7	Low
Natarajan, K. et al., 2022 [28]	1	1	1	1	2	1	1	1	9	Low
Jara, A. et al., 2022 [29]	1	1	1	1	2	1	1	1	9	Low
Menni, C. et al., 2022 [32]	1	1	1	1	2	1	1	1	9	Low
Khong, K. W. et al., 2022 [33]	1	1	1	1	2	1	1	1	9	Low
Çağlayan, D. et al., 2022 [34]	1	1	1	1	2	1	1	1	9	Low
Angkasekwinai, N. et al., 2022 [38]	1	1	1	1	1	1	1	1	8	Low
Mok, C. K. P. et al., 2022 [39]	1	1	1	1	2	1	1	1	9	Low
Starrfelt, J. et al., 2022 [36]	1	1	1	1	2	1	1	1	9	Low
Baum, U. et al., 2022 [44]	1	1	1	1	2	1	1	1	9	Low

**Table 5 ijerph-19-10752-t005:** Risk of bias and quality assessment by Newcastle-Ottawa quality assessment scale (NOS) of the included case-control studies.

First Author (Published Time)	Selection				Comparability	Exposure			Number of Stars	Risk of Bias
	Is the Case Definition Adequate?	Representativeness of the Cases	Selection of Controls	Definition of Controls	Comparability of Cases and Controls on the Basis of the Design or Analysis	Ascertainment of Exposure	Same Method of Ascertainment for Cases and Controls	Non-Response Rate		
Ranzani, Otavio T. et al., 2022 [42]	1	1	1	1	2	1	1	0	8	Low
Andrews, N. et al., 2022 [43]	1	1	1	1	2	1	1	0	8	Low

**Figure 2 ijerph-19-10752-f002:**
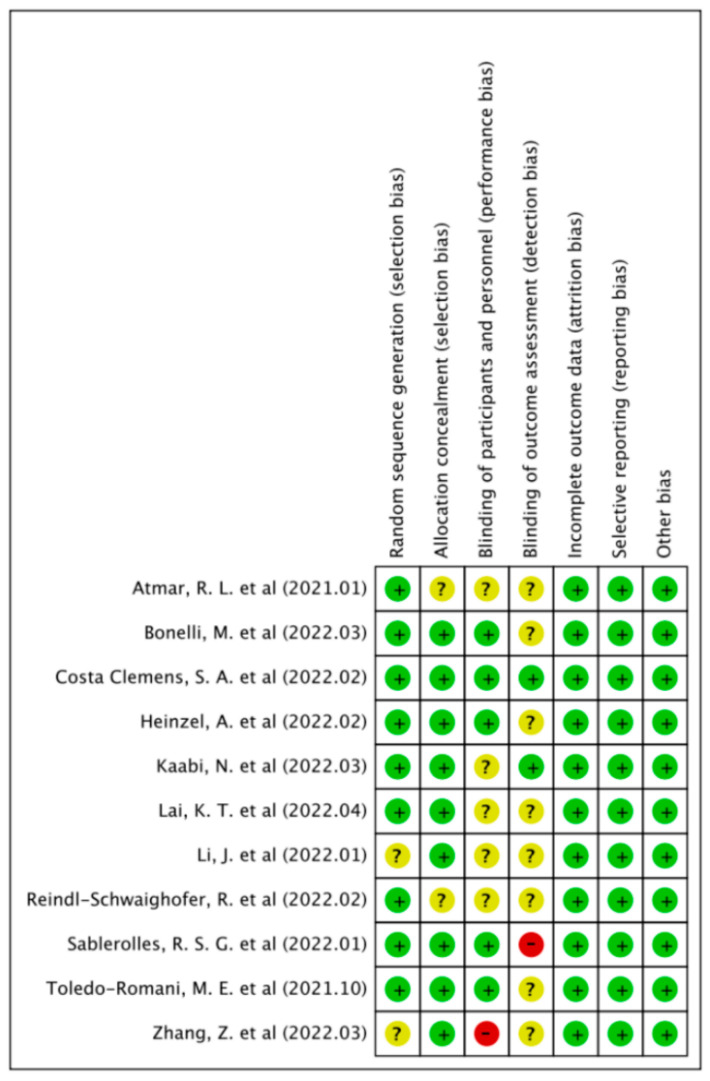
Risk of bias assessment of the included RCTs [14,24,26,27,30,31,35,37,40,41,45].

## 4. Discussion

For now, vaccination campaigns remain key to mitigating the wave of COVID-19, helping to prevent COVID-19-related severe illness and death, and helping to prevent health burden events, such as long-COVID-19 sequelae. However, after a period of vaccination, there are some weakened immunities, but there are protective effects, such as the booster vaccination can provide a double-growth decline in antibodies and can better deal with mutant strains. Therefore, it is of great significance to strengthen immunization for key and susceptible populations. Heterologous vaccination has become as important as homologous vaccination in enhancing immunization. The difference of safety and effectiveness between the two has attracted people’s attention.

Our systematic review and meta-analysis of 23 studies, involving 1,726,506 inoculation cases of homologous booster doses and 5,343,580 inoculation cases of heterologous booster doses, provided VE for homologous and heterologous vaccines against adverse outcomes of COVID-19 and performed a subgroup analysis to compare the differences based on prime-booster regime, study design, and article type. At the same time, we also provided the OR of safety events of homologous and heterologous vaccination groups, as well as the comparison of immunogenicity. Different from previous study that compared the effectiveness of homologous and heterologous booster for SARS-CoV-2 vaccines [13], we also focused on safety and immunogenicity and conducted a more in-depth study on the differences between the two groups.

Available data showed that the homologous and heterologous boosters of SARS-CoV-2 vaccines provided a high level of protection against COVID-19 for individuals with a complete primary regimen for COVID-19 vaccine. Heterologous booster showed higher VE than homologous booster vaccination in the prevention of SARS-CoV-2 infection (96.1% vs. 84%), symptomatic COVID-19 (56.8% vs. 17.3%), and COVID-19 hospital admissions (97.4% vs. 93.2%). We observed that a mRNA booster or a recombinant booster after two inactivated vaccines both showed superior effectiveness over other heterologous vaccination regimes in preventing COVID-19 admissions, COVID-19 ICU admissions, and COVID-19-related death, with VE higher than 95%. Our results was consistent with those of a large-scale prospective cohort study [29], it showed that the VE in preventing symptomatic COVID-19 was 78.8% for the homologous booster group (three-dose CoronaVac), 96.5% for primary series (two-dose CoronaVac) with a BNT162b2 booster, and 93.2% for primary series with a AZD1222 booster. Additionally, the VE in preventing COVID-19 hospital admissions, COVID-19 ICU admissions, and COVID-19-related death was 86.3%, 92.2%, and 86.7% for the homologous booster, 96.1%, 96.2%, and 96.8% for a BNT162b2 booster, and 97.7%, 98.9%, and 98.1% for an AZD1222 booster. For all outcomes, a heterologous booster showed higher VE than a homologous booster.

The high VE by the heterologous booster may be related to levels of antibody titer and T-cell responses. A study of Turkish healthcare workers showed, in the heterologous booster group (two-dose CoronaVac with a BNT162b2 booster), median antibody levels at four months after the second dose of CoronaVac and at 7–67 days after third dose were 168 AU/mL and 17,609 AU/mL, respectively, with a 104.8-fold increase; for the homologous booster group (three-dose CoronaVac), the median antibody levels were 141.1 AU/mL and 1237.9 AU/mL, respectively, with an 8.7-fold increase. Compared with homologous booster group, a 14.2-fold increase was detected in the BNT162b2 booster group [34]. The results of a phase 4 randomized trial showed that neutralizing antibody against wild-type SARS-CoV-2 was from 2.5 at 0 day before booster vaccination to 197.4 at 14 days after vaccination in the heterologous booster group (two-dose CoronaVac with a Concidecia booster) and from 2.2 at 0 day before booster vaccination to 33.6 at 14 days after vaccination in the homologous booster group (three-dose CoronaVac). Although GMTs of neutralizing antibody was decreased slightly in both groups at 28 days after boosting, in the heterologous booster group it was still significantly higher than that in the homologous booster group [14]. Our study showed that the heterologous booster group has not only a higher level of neutralizing antibody but also a higher level of T-cell response rate than that in the homologous booster group, which was consistent with the results of Roos S G Sablerolles et al. They assessed the T-cell response based on the levels of Interferon-γ and found the heterologous booster group (one-dose Ad26.COV2.S with a mRNA-1273 or BNT162b2 booster) induced higher levels of T-cell response than the homologous group (91.7%, 91.5% vs. 72.7%) [30]. Available data showed that both the homologous and heterologous boosters induced systemic and injection site adverse reactions. The heterologous booster group had a higher risk of partial systemic adverse reactions. No serious adverse events were observed in the homologous and heterologous booster groups in our study. Previous studies showed that these adverse events could be resolved in short period [30,46]. Toshio Naito found that most adverse reactions were mild to moderate and could be resolved within one week with no sequelae in people who received a BNT162b2 or mRNA-1273 COVID-19 booster vaccinations after two doses of BNT162b2 [46]. In addition, the incidence of adverse reactions tended to decrease in those ≥60 years [46]. Roos S.G. Sablerolles found that adverse events within 7 days after booster vaccination were mild to moderate, with symptoms generally resolved within 48 h [30].

### Differences in Effectiveness between Homologous and Heterologous Boosters for SARS-CoV-2 Variants

The VE was higher in a complete primary regimen with a booster group in preventing COVID-19 hospital admissions caused by Omicron variants than a primary regimen only group [47]. A study from Israel showed that a third dose of the mRNA vaccine significantly reduced the rate of confirmed SARS-CoV-2 infection and severe COVID-19 illness [48]. Numerous studies also provided clear serological evidence. Vaccination with a booster can induce the level of antibody concentrations against different variants of SARS-CoV-2, and optimum concentration of antibodies can provide significant protection against infection from variants [49]. A study conducted among people who were aged ≥18 years with a previous positive nucleic acid amplification test (NAAT) or diagnosis of COVID-19 found the VE in preventing COVID-19 hospital admissions was 57.8% during Delta-predominant period and 67.6% during Omicron-predominant period [50]. In a slightly different way, the results of another study showed a higher effectiveness after booster vaccination during the Delta period compared with the Omicron period (70% vs. 54%) [51]. A previous study showed that both the homologous booster and the heterologous boosters have better protection against COVID-19 hospitalizations caused by Omicron variant where VE was 77% in mRNA1273 homologous series and 30% in AD25. In COV2 homologous series, VE was 64% in the heterologous booster group (one-dose Johnson with a mRNA vaccine booster) [47]. A systematic review showed that a third dose of vaccine led to a significant increase in serum neutralization of Omicron variants in both heterologous and homologous booster groups. Neutralizing titers against the Omicron variant increased by 1.17 to 96.94 folds in the homologous booster group, while the wild-type, alpha, beta, and delta variants increased by 1.85 to 53.83, 3.15, 2.94 to 119.84, and 1.80 to 53.81, respectively. Similarly, neutralization titers for Omicron variants, wild-type, alpha, beta, and delta variants were found to increase by 2 to 15.87, 18.4 to 36.1, 17.4 to 89.22, and 27.9 to 42.80, respectively, in the heterologous booster group [52]. Additionally, Ai and his colleagues found that Omicron might more likely escape vaccine-induced immune protection after a third heterologous booster of protein subunit vaccine (ZF2001) compared to prototypes and other variants of concern [25].

At present, vaccination is still the key to control the epidemic. Booster vaccination can help prevent COVID-19-related severe illness and death and has good protection against different SARS-CoV-2 variants. Heterologous boosted vaccination provides an alternative to enhanced immunogenicity and is a process of improving immunization strategies. There are still few original studies on the effectiveness and safety of homologous and heterologous boosters of SARS-CoV-2 vaccines, and more relevant evidence is needed in the future to continuously improve immunization strategies.

Our study has some limitations. First, the small number, different study designs, and different time frames of included articles in which vaccine effectiveness was calculated, resulted in large confidence intervals and lack of accuracy of some outcomes. Second, the I^2^ of some outcomes were large, which may be caused by the difference in age and sex ratio of the study population. Third, although available data showed that the heterologous booster group had stronger immunogenicity than that of the homologous vaccinated group, due to limited data, the GMTs level of only part of the variants was analyzed. Fourth, we analyzed data 14 days after the booster vaccination mainly, data for longer periods, such as 28 days after booster vaccination or longer, were limited, which could not fully explain the difference in immunogenicity between the homologous and heterologous vaccination groups. Fifth, in the original study design, we sought to perform subgroup analysis of the effectiveness of the homologous and heterologous boosters of different age group recipients, but due to the limited data, we could not do so. Sixth, a significant number of the included participants were primarily immunized with two doses of inactivated vaccine and boosted with mRNA or other technology; however, evidence on advantages in terms of effectiveness for other regimens was still limited. For the third limitation, we made some supplements in the discussion.

## 5. Conclusions

Our results suggested homologous and heterologous booster of SARS-CoV-2 vaccines provided high VE in preventing SARS-CoV-2 infection, COVID-19 hospital admissions, COVID-19 ICU admissions, and COVID-19-related death. The heterologous booster showed higher VE than the homologous booster in the prevention of SARS-CoV-2 infection, symptomatic COVID-19, and COVID-19 hospital admissions. Compared with the homologous booster group, there was a higher risk of fever, myalgia, and malaise or fatigue within 7 days after boosting and a higher risk of malaise or fatigue within 28 days after boosting in the heterologous booster group. Our findings suggested that both the homologous and heterologous COVID-19 booster doses had great effectiveness, immunogenicity, and acceptable safety, and a heterologous booster dose was more effective, which would help make appropriate public health decisions, reduce public hesitancy in vaccination, and promote rational allocation of COVID-19 vaccine resources.

## Figures and Tables

**Figure 1 ijerph-19-10752-f001:**
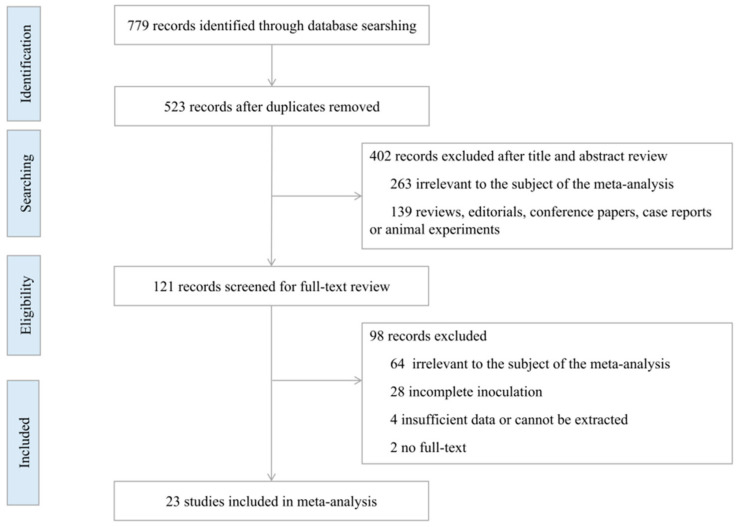
Flowchart of the study selection.

**Table 1 ijerph-19-10752-t001:** Comparison of safety between heterologous booster and homologous booster.

Safety Events	Number of Data Source	Heterologous Booster (n/N)	Homologous Booster (n/N)	OR	95% CI	*p*-Value	Weight (%)	I2	P-Heterogeneity
Systematic Adverse Reaction (within 7 Days after Boosting)									
Systematic adverse reaction	2	42029/155484	2074/142400	6.02	0.34–107.31	>0.05	100	99.70%	<0.05
Malaise or fatigue	13	677/1913	421/1645	1.745	1.047–2.906	<0.05	100	79.70%	<0.05
Chills	9	455/1737	182/1155	1.428	0.808–2.524	>0.05	100	82.60%	<0.05
Fever	15	351/2861	126/2227	1.930	1.199–3.107	<0.05	100	59.30%	<0.05
Headache	15	893/2861	454/2227	1.418	0.940–2.140	>0.05	100	81.50%	<0.05
Nausea	11	159/1775	100/1536	1.114	0.788–1.574	>0.05	100	16.30%	>0.05
Vomiting	3	2/945	1/965	2.02	0.183–22.318	>0.05	100	-	-
Abdominal pain	2	2/169	0/128	2.518	0.258–24.563	>0.05	100	0.00%	>0.05
Diarrhea	7	34/1253	16/1160	1.522	0.758–3.055	>0.05	100	0	>0.05
Arthralgia	8	179/1685	123/1477	1.055	0.644–1.730	>0.05	100	55.30%	<0.05
Myalgia	15	884/2861	401/2227	1.825	1.079–3.089	<0.05	100	88.10%	<0.05
Skin rash	2	5/138	2/109	1.734	0.329–9.144	>0.05	100	-	-
Dizziness	2	7/936	4/944	1.597	0.261–9.786	>0.05	100	38.10%	>0.05
Serious adverse events (SAE)	2	0/145	0/119	-	-	-	-	-	-
Systematic Adverse Reaction (within 28 Days after Boosting)									
Arthralgia	2	14/123	9/130	2.084	0.741–5.866	>0.05	100	0.00%	>0.05
Malaise or fatigue	2	32/123	16/130	4.140	1.729–9.916	<0.05	100	0.00%	>0.05
Myalgia	2	16/123	9/130	2.694	0.953–7.614	>0.05	100	0.00%	>0.05
Injection Site Adverse Reaction (within 7 Days after Boosting)									
Injection site adverse reaction	6	125599/155621	102419/142630	1.053	0.535–2.070	>0.05	100	84.80%	<0.05
Erythema or redness	7	48/1556	47/1391	0.806	0.420–1.548	>0.05	100	37.60%	>0.05
Induration or swelling	7	93/1556	79/1389	0.802	0.385–1.670	>0.05	100	65.90%	<0.05

**Table 2 ijerph-19-10752-t002:** Comparison of immunogenicity between heterologous booster and homologous booster.

Index	Number of Study	Heterologous Booster (n/N)	Homologous Booster (n/N)	Effect (SMD)	95% CI	*p*-Value		Weight (%)	I2	P-Heterogeneity	
GMT of Neutralizing Antibody for VOCs											
Anti-RBD IgG (14 days After Boosting)	8	1124	1067	1.244	0.900–1.588	0	<0.05	100	90.90%	0	<0.05
GMT of Neutralizing Antibody for VOCs											
Anti-wild-type (14 days after boosting)	4	295	303	1.028	0.654–1.402	0	<0.05	100	77.90%	0.004	<0.05
Anti-wild-type (28 days after boosting)	4	206	212	0.967	0.571–1.363	0	<0.05	100	67.90%	0.025	<0.05
Anti-Delta (14 days after boosting)	4	295	303	0.833	0.509–1.157	0	<0.05	100	71.80%	0.014	<0.05
**Index**	**Number of Data Source**	**Heterologous Booster (n/N)**	**Homologous Booster (n/N)**	**OR**	**95% CI**	***p*-Value**		**Weight (%)**	**I2**	**P-Heterogeneity**	
Response Rate of Antibody											
response rate of anti-spike IgG (28 days after boosting)	2	264/321	115/205	5.536	2.738–11.195	0	<0.05	100%	0	0.495	>0.05
response rate of neutralizing antibody (nearly 1 month after boosting)	4	160/289	93/234	1.446	0.864–2.422	0.161	>0.05	100%	0	0.49	>0.05
Response Rate of T cell											
response rate of gama interferon (14 days after boosting in patients who were negative in baseline)	4	54/87	2/18	16.3	3.439–77.272	0	<0.05	100%	0	0.321	>0.05

**Table 3 ijerph-19-10752-t003:** Comparison of effectiveness between heterologous booster and homologous booster.

Homologous Booster Dose								
Outcomes		No. of Studies	Vaccine Effectiveness (%)	95% CI	I2	P-Heterogeneity	*p* Value for Subgroup Differences	Weight (%)
Prevention of SARS-CoV-2 Infection								
Article type	Published	3	85.80%	84.3–87.2%	98.30%	0	0	66.51
	Preprint	5	80.50%	78.4–82.5%	92.90%	0	0	33.49
Study design	Cohort study	7	83.30%	82.1–84.5%	96.60%	0	0	92.84
	RCT	1	92.40%	88.1–96.7%	-	-	0	7.16
Immune condition	mRNA vaccine ×3	6	86.40%	84.8–88.0%	96.50%	0	0	55.29
	inactivated vaccine ×3	1	78.80%	76.9–80.7%	-	-	0	37.55
	SOBERANA ×2 + SOBERANA plus ×1	1	92.40%	88.1–96.7%	-	-	0	7.16
Overall		8	84.00%	82.8–85.1%	96.40%	0		100
Prevention of Symptomatic COVID-19								
Article type	Published	1	75.50%	60.4–90.6%	-	-	0	96.2
	Preprint	1	15%	12–18%	-	-	0	3.8
Overall		2	17.30%	14.4–20.2%	98.30%	0	0	100
Prevention of Emergency Department and Urgent Care Admissions								
Immune condition	mRNA vaccine ×3	1	83.00%	82.0–83.7%	-	-	0	99.1
	adenovirus vector vaccine ×2	1	54.00%	44.0–64.0%	-	-	0	0.99
Overall		2	82.70%	81.7–83.7%	96.90%	0	0	100
Prevention of COVID-19 Hospital Admissions								
Article type	published	3	88.70%	87.5–90.0%	89.10%	0	0	38.44
	preprint	1	93.20%	92.4–94.0%	-	-	0	61.56
Immune condition	mRNA vaccine ×3	2	94.20%	93.3–95.0%	97.60%	0	0	88.92
	inactivated vaccine ×3	1	86.30%	83.9–88.7	-	-	0	10.69
	adenovirus vector vaccine ×2	1	67.00%	54.5–79.5%	-	-	0	0.39
Overall		4	93.20%	92.4–94.0%	96.90%	0	0	100
**Heterologous Booster Dose**								
**Outcomes**		**No. of Studies**	**Vaccine Effectiveness (%)**	**95% CI**	**I2**	**P-Heterogeneity**	***p*-Value for Subgroup Differences**	**Weight (%)**
Prevention of SARS-CoV-2 Infection								
Immune condition	adenovirus vector vaccine ×1 + mRNA vaccine ×1	2	90.70%	89.1–92.2%	5.60%	0.303	0	2.42
	inactivated vaccine ×2 + recombinant vaccine ×1	1	93.20%	92.4–94.0%	-	-	0	8.68
	inactivated vaccine ×2 + mRNA vaccine ×1	1	96.50%	96.2–96.7%	-	-	0	88.9
Overall		4	96.10%	95.8%-96.3%	97.30%	0	0	100
Prevention of Symptomatic COVID-19								
Article type	Published	1	71.40%	49.3–93.5%	-	-	0	0.06
	Preprint	1	56.80%	56.3–57.4%	-	-	0	99.94
Immune condition	adenovirus vector vaccine ×2 + mRNA vaccine ×1	1	71.40%	49.3–93.5%	-	-	0	0.06
	inactivated vaccine ×2 + mRNA vaccine ×1	1	56.80%	56.3–57.4%	-	-	0	99.94
Overall		2	56.80%	56.3–57.4%	40.30%	0.196	0	100
Prevention of COVID-19 Hospital Admissions								
Immune condition	adenovirus vector vaccine ×1 + mRNA vaccine ×1	1	78.00%	71.0–85.0%	-	-	0	0.21
	inactivated vaccine ×2 + mRNA vaccine ×1	1	96.10%	96.3–96.9%	-	-	0	16.03
	inactivated vaccine ×2 + recombinant vaccine ×1	1	97.40%	97.3–98.1%	-	-	0	83.76
Overall		3	97.40%	97.1–97.7%	95.30%	0	0	100
Prevention of COVID-19 ICU Admissions								
Immune condition	inactivated vaccine ×2 + mRNA vaccine ×1	1	96.20%	94.9–97.6%	-	-	0	6.3
	inactivated vaccine ×2 + recombinant vaccine ×1	1	98.90%	98.6–99.3%	-	-	0	93.7
Overall		2	98.70%	98.4–99.1%	93.10%	0	0	100
Prevention of COVID-19-Related Death								
Immune condition	inactivated vaccine ×2 + mRNA vaccine ×1	1	96.80%	94.5–99.1%	-	-	0	7.7
	inactivated vaccine ×2 + recombinant vaccine ×1	1	98.10%	97.5–98.8%	-	-	0	92.3
Overall		2	98.00%	97.4–98.6%	15.50%	1.277	0	100

## Data Availability

Data are available from the corresponding author by request.

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
