# Peer review of "Comparison of the Effectiveness and Safety of Heterologous Booster Doses with Homologous Booster Doses for SARS-CoV-2 Vaccines: A Systematic Review and Meta-Analysis"

_ijerph, 2022, doi:10.3390/ijerph191710752_

Round 1
Reviewer 1 Report
The review by Jie Deng et al demonstrates the comparison between the heterologous booster doses and homologous booster doses for COVID-19 vaccines in terms of effectiveness, immunogenicity, and safety. It is a well-written analytical review and could be of decent importance in the real world in making public health decisions regarding the COVID-19 vaccine immunizations.
Did the authors analyze the homologous and heterologous booster effectiveness in different populations? It would be interesting to see if different age group recipients have different effects of the vaccine.
Authors should consider including more information about different COVID-19 vaccines (types, doses) used in the selected 23 studies for meta-analysis.
Did the authors observe superior effectiveness by any one of the heterologous vaccination regimes over others?
Authors should consider revising the review and correcting the grammatical errors. Some are outlined here. Typo in lines 290, 333. Avoid repeated use of one word in one sentence like ‘weakened’ in line 267. In line 290, the authors state ‘In that study…’ Mention the article being referred to here. Rephrase lines 305-309, 318-321 for better clarity. Instead of beginning a paragraph with ‘ This may be related to…’ in line 298, connect it with the previous paragraph, for instance ‘ The high VE by heterologous booster may be related to…’
Cite the reference in lines 321-324.
Author Response
Did the authors analyze the homologous and heterologous booster effectiveness in different populations? It would be interesting to see if different age group recipients have different effects of the vaccine.
Response: Thanks for the reviewer’s suggestion. In the original study design, we sought to perform subgroup analysis of the effectiveness of the homologous and heterologous booster of different age group recipients, but due to the limited data, we did not do so. And we have added it to the limitations.
Authors should consider including more information about different COVID-19 vaccines (types, doses) used in the selected 23 studies for meta-analysis.
Response: Thanks for the reviewer’s suggestion. Information about types and doses of different COVID-19 vaccines was extracted and listed in ‘Supplementary table 1. Characteristics of the included studies’ which was uploaded as a supplementary material originally.
Did the authors observe superior effectiveness by any one of the heterologous vaccination regimes over others?
Response: Thanks for the reviewer’s suggestion. We observed that a mRNA booster or a recombinant booster after two inactivated vaccines both showed superior effectiveness over other heterologous vaccination regimes in preventing COVID-19 admissions, COVID-19 ICU admissions and COVID-19 related death, with VE higher than 95%. And We have added some discussion in discussion section.
Authors should consider revising the review and correcting the grammatical errors. Some are outlined here. Typo in lines 290, 333. Avoid repeated use of one word in one sentence like ‘weakened’ in line 267. In line 290, the authors state ‘In that study…’ Mention the article being referred to here. Rephrase lines 305-309, 318-321 for better clarity. Instead of beginning a paragraph with ‘ This may be related to…’ in line 298, connect it with the previous paragraph, for instance ‘ The high VE by heterologous booster may be related to…’
Response: Thanks for the reviewer’s suggestion. We have revised the wrong grammar and inappropriate statement, see in lines 280, lines 287-291, lines 300-303, line 323, etc.
Cite the reference in lines 321-324.
Response: Thanks for the reviewer’s suggestion. We have added the cited reference.
Reviewer 2 Report
Dear authors,
Your work intends to carry out a systematic review and meta-analysis on an important topic, at a sensitive time, when decisions need to be made by health authorities on how the next vaccine boosters should be programmed.
I think that idea that heterologous vaccine booster doses are more effective is biased from the start, as in most studies, the prime immune condition of the participants is based on a 2-dose of inactivated vaccines, and in In a large number of these 23 studies, the booster vaccine is based on mRNA or other technology. Considering that inactivated vaccines offer lower protection than other vaccine technologies the heterologous vaccination can only have advantages in terms of effectiveness. Please consider placing more weight on studies where these conditions are not met.
The 23 studies include very different study designs and different time frames in which vaccine effectiveness was calculated. You refer to these aspects as the limitations of your study. I think it was better to do only a systematic review.
In the discussion of this manuscript, you mention reference 46 (RECORD Nº 1 at the supplementary table). “For all outcomes, a heterologous booster showed higher VE than a homologous booster.” and you emphasize these results to accomplish that VE is higher with a heterologous booster. This reference is repeated in Ref. 28? (Meta-analysis: references 14, 23-44”?)
Kind regards.
Author Response
I think that idea that heterologous vaccine booster doses are more effective is biased from the start, as in most studies, the prime immune condition of the participants is based on a 2-dose of inactivated vaccines, and in In a large number of these 23 studies, the booster vaccine is based on mRNA or other technology. Considering that inactivated vaccines offer lower protection than other vaccine technologies the heterologous vaccination can only have advantages in terms of effectiveness. Please consider placing more weight on studies where these conditions are not met.
Response: Thanks for the reviewer’s suggestion. We also agreed that it should consider placing more weight on studies where these conditions were not met and we have added the relevant content to the discussion section according to your suggestion.
The 23 studies include very different study designs and different time frames in which vaccine effectiveness was calculated. You refer to these aspects as the limitations of your study. I think it was better to do only a systematic review.
Response: Thanks for the reviewer’s suggestion. We have conducted subgroup analyses for different study designs and time frames, as shown in Table 3. Besides, we have added this limitation in the discussion section according to your suggestion.
In the discussion of this manuscript, you mention reference 46 (RECORD Nº 1 at the supplementary table). “For all outcomes, a heterologous booster showed higher VE than a homologous booster.” and you emphasize these results to accomplish that VE is higher with a heterologous booster. This reference is repeated in Ref. 28? (Meta-analysis: references 14, 23-44”?)
Response: Thanks for the reviewer’s suggestion. References 28 and 46 were indeed duplicates, and we have deleted Ref.46 and adjusted the corresponding reference serial number.
Reviewer 3 Report
The authors have carried out an important research and presented in a well defined way. As we all know the implications of booster doses as these can provide significant amount of protection against SARS-CoV-2 and its emerging variants.
However, My major question is in the conclusion it is saying that both heterologous and homologous COVID-19 booster doses had great effectiveness, immunogenicity. I personaly think that many studies have shown the advantages of heterologous vaccination over the homologous.
Therefore, I think the conclusion must be elaborated while describing or highlighting the advantages of the heterologous vaccination.
Secondly, In the discussion or in the introduction the implications or advantages of booster doses of vaccines against novel variants of SARS-CoV-2 can be highlighted.
Similarly, few lines can be incorporated about the booster doses and variants in the conclusion section.
The following papers can be utilized
Mohapatra RK, El-Shall NA, Tiwari R, Nainu F, Kandi V, Sarangi AK, Mohammed TA, Desingu PA, Chakraborty C, Dhama K. Need of booster vaccine doses to counteract the emergence of SARS-CoV-2 variants in the context of the Omicron variant and increasing COVID-19 cases: An update. Hum Vaccin Immunother. 2022 Nov 30;18(5):2065824. doi: 10.1080/21645515.2022.2065824. Epub 2022 May 20. PMID: 35594528.
Dhawan M, Emran TB, Choudhary OP. Implications of COVID-19 vaccine boosters amid the emergence of novel variants of SARS-CoV-2. Ann Med Surg (Lond). 2022 May;77:103612. doi: 10.1016/j.amsu.2022.103612. Epub 2022 Apr 12. PMID: 35432993; PMCID: PMC9001012.
Stefanelli P, Rezza G. COVID-19 Vaccination Strategies and Their Adaptation to the Emergence of SARS-CoV-2 Variants. Vaccines (Basel). 2022 Jun 6;10(6):905. doi: 10.3390/vaccines10060905. PMID: 35746513; PMCID: PMC9229267.
Author Response
However, My major question is in the conclusion it is saying that both heterologous and homologous COVID-19 booster doses had great effectiveness, immunogenicity. I personaly think that many studies have shown the advantages of heterologous vaccination over the homologous.Therefore, I think the conclusion must be elaborated while describing or highlighting the advantages of the heterologous vaccination.
Response: Thanks for the reviewer’s suggestion. We have added some description of the advantages of the heterologous booster vaccination in conclusion section.
Secondly, In the discussion or in the introduction the implications or advantages of booster doses of vaccines against novel variants of SARS-CoV-2 can be highlighted. Similarly, few lines can be incorporated about the booster doses and variants in the conclusion section.
Response: Thanks for the reviewer’s suggestion. We have added some discussion about advantages of booster doses of vaccines against novel variants of SARS-CoV-2 in 4.1 section. See in lines 343-348, A study from Israel showed that a third dose of the mRNA vaccine significantly reduced the rate of confirmed SARS-CoV-2 infection and severe COVID-19 illness. Numerous studies also provided clear serological evidence. Vaccination with a booster can induce the level of antibody concentrations against different variants of SARS-CoV-2, and optimum concentraion of antibodies can provide significant protection against infection of variants. And see in lines 365-368, besides, Ai and his colleagues found that Omicron might more likely escaped vaccine-induced immune protection after a third heterologous booster of protein subunit vaccine (ZF2001) compared to prototypes and other variants of concern.
The following papers can be utilized .
Mohapatra RK, El-Shall NA, Tiwari R, Nainu F, Kandi V, Sarangi AK, Mohammed TA, Desingu PA, Chakraborty C, Dhama K. Need of booster vaccine doses to counteract the emergence of SARS-CoV-2 variants in the context of the Omicron variant and increasing COVID-19 cases: An update. Hum Vaccin Immunother. 2022 Nov 30;18(5):2065824. doi: 10.1080/21645515.2022.2065824. Epub 2022 May 20. PMID: 35594528.
Dhawan M, Emran TB, Choudhary OP. Implications of COVID-19 vaccine boosters amid the emergence of novel variants of SARS-CoV-2. Ann Med Surg (Lond). 2022 May;77:103612. doi: 10.1016/j.amsu.2022.103612. Epub 2022 Apr 12. PMID: 35432993; PMCID: PMC9001012.
Stefanelli P, Rezza G. COVID-19 Vaccination Strategies and Their Adaptation to the Emergence of SARS-CoV-2 Variants. Vaccines (Basel). 2022 Jun 6;10(6):905. doi: 10.3390/vaccines10060905. PMID: 35746513; PMCID: PMC9229267.
Response: Thanks for the reviewer’s suggestion. We have added some discussion about the advantages of booster dose against the SARS-CoV-2 variants in the discussion section.